# Flight Controller as a Low-Cost IMU Sensor for Human Motion Measurement

**DOI:** 10.3390/s23042342

**Published:** 2023-02-20

**Authors:** Artur Iluk

**Affiliations:** Faculty of Mechanical Engineering, Wroclaw University of Science and Technology, 50-370 Wroclaw, Poland; artur.iluk@pwr.edu.pl

**Keywords:** human body pose estimation, IMU sensors, motion capture, flight controller

## Abstract

Human motion analysis requires information about the position and orientation of different parts of the human body over time. Widely used are optical methods such as the VICON system and sets of wired and wireless IMU sensors to estimate absolute orientation angles of extremities (Xsens). Both methods require expensive measurement devices and have disadvantages such as the limited rate of position and angle acquisition. In the paper, the adaptation of the drone flight controller was proposed as a low-cost and relatively high-performance device for the human body pose estimation and acceleration measurements. The test setup with the use of flight controllers was described and the efficiency of the flight controller sensor was compared with commercial sensors. The practical usability of sensors in human motion measurement was presented. The issues related to the dynamic response of IMU-based sensors during acceleration measurement were discussed.

## 1. Introduction

Human motion analysis is required in many scientific activities, such as performance analysis in different sports disciplines [1] and biomechanical research in the case of musculoskeletal dysfunctions [2]. Measurement of angles and position of the different parts of the human body allows an indirect estimation of the forces that act in the joints and the active forces applied by the muscles (open sim). For the estimation of the actual pose of the body, the most critical is the history of the angle of the joints, because all other parameters such as, for example, the actual distances between the joints, can be extracted from the angle analysis in a simple calibration procedure [3]. Linear and angular velocities and acceleration can also be calculated from a known history of joint angles.

In general, two types of system are used for human motion capture. The optical systems are based on passive or active optical markers located on the human body and a set of cameras observing the markers during the motion [4]; however, some systems are evolving in the direction of an optical markerless system at the cost of decreased accuracy [1]. Image processing software allows identification of the absolute position of each marker that is in the field of view of at least two cameras. With knowledge of the camera position and orientation, crossing optical lines between cameras and marker determines the actual position in 3D space [5]. An example of an optical tracking system is VICON [6,7].

The advantage of optical systems is the absolute position of each identified marker in space and the low weight of the markers, which does not influence motion. However, this kind of tracker system requires quite expensive equipment, high-resolution cameras, good quality optics with low optical distortion, and high-performance computers to analyze the motion, and last but not least, expensive software. Another disadvantage is, especially in tracking complex motion, the requirement for continuous visibility of markers. In order to avoid covering of some markers by the moving body, many cameras are necessary.

The second group of motion tracking devices are sensors based on Inertial Measurement Units (IMUs) [2,8,9]. This kind of device uses a set of three gyroscopes and three accelerometers to estimate the actual sensor attitude (3D orientation) in space during dynamic motion. The main disadvantage of the IMU sensor is the measurement of relative motion rather than absolute position, and the consequence is the so-called drift of orientation. The signals from accelerometers and gyroscopes are integrated in time, and this process results in a cumulative error of estimation. After a long time, the IMU sensor can report some movement even if the sensor is not moving. To minimize the increasing error of the attitude, advanced algorithms such as Kalman filtering are used to fuse redundant information from different sensors and minimize the drift. In some applications, the set of three magnetometers is also used for a better estimation of the direction of the sensor in relation to the magnetic field of the Earth, but in practice it can only be used in a perfectly stable magnetic environment due to high sensitivity to any artificial source of magnetic field [10,11,12].

Another problem is the mass of sensors. Even if modern IMU sensors are miniaturized, it requires power, and the data need to be stored locally or transmitted to the external devices. To fully resemble body motion, a set of 10 to 17 sensors should be used [3,8,13]. The size and mass of sensors in some application can influence motion. In addition, wireless sensors must be synchronized to provide correct joint angles. Miniaturized sets of wireless sensors with a high rate of measurement are quite expensive.

The comparison of the specific features of IMU-based and optical methods of human body movement techniques was provided in [7]. The conclusion was that the optical systems provide a better reaction time than the IMU-based systems. The example view of both types of systems is presented in Figure 1.

In the paper, the novel concept of a low-cost measurement system based on COTS flight controllers with custom software is presented. In the existing literature there is no description of use of the flight controller as a human body motion sensor. The usage of a synchronized set of flight controllers was presented and the device was preliminary tested.

The method used for the synchronization of multiple flight controller sensors was described. The angle estimation performance of the flight controller sensor was compared with two types of commercial sensors.

With Xsens MTi G-700, a much more precise instrument, used as a reference sensor to check the accuracy of the flight controller sensor;With MTw Awinda sensor, designed especially for human body motion measurements, to check the possibility of its replacement by a much cheaper flight controller sensor.

Additionally, a simple test of the human body motion during walking was carried out with the use of this type of sensor to present its ability to capture highly dynamic events. Issues related to the synchronization with external sensors and dynamic response of IMU-based sensors were presented and discussed.

## 2. Methods

In the case of measuring human body movement with the use of MEMS sensors, the main three configurations are used [14,15]:Local powering and recording: sensors are powered by an internal battery located inside the sensor and data are recorded in internal memory. The main problems are the mass of the sensor, the synchronization of the recording between sensors, and the limited time of the recording;Local powering and remote wireless recording: sensors are powered by an internal battery, and the stream of data are transmitted wirelessly to the stationary recorder. Synchronization of the recording is easier, but the main problem is the available bandwidth to send high-frequency data from many sensors simultaneously;Cable powering and recording: All sensors are powered by a single battery located in the central unit located on the body; data are transferred through the cable to the recorder in the central unit. The data cables should provide high transfer capacity, the central unit is relatively heavy.

The most popular systems with local powering and remote recording are MTw Awinda (Xsens) and Ultium (Noraxon). The sensors of these systems are shown in Figure 2. Examples of orientation and acceleration sensors equipped with a microprocessor that also calculates the actual orientation but with cable powering and recording are presented in Figure 3.

In the setup presented in the paper, a different approach was used. The concept is based on the following.

Local recording in the sensor;Central powering by cable;Central triggering by the same cable, providing good synchronization between the sensors.

As sensor hardware, the standard flight controller unit was adopted. Flight controllers [16], used in various types of drone, are miniaturized devices that contain:IMU sensor with a set of gyroscopes and accelerometers;Microprocessor used for real-time data processing and to estimate actual orientation of the controller in space;Additional internal memory to store the flight parameters, the so-called Black Box;A set of serial interfaces that can be used for bidirectional communication with other devices.

Flight controllers are available in many versions, but the biggest advantage is open source software supported by the drone users community, such as Cleaflight, Betaflight, and iNav [17]. The example of a low-cost flight controller is presented in Figure 4. It is a clone of the Omnibus F4 Pro flight controller that works under the control of the Betaflight software. The size of the controller is 36 × 36 × 8 mm, with a mass of 7 g. For comparison, the size of the AWINDA MTw sensor with local powering and remote recording is 47 × 30 × 13 mm and the mass is 16 g.

The controller is equipped with three UART digital ports to communicate with various external components (Figure 4). One of them allows us to connect to wireless receiver, which is used to trigger and stop the recording process.

On the other side of the controller there is an SD card slot, which can be used to store the recorded data (Figure 5). An SD card with a capacity of 32 GB can be used, which provides an extremely long recording time. Due to the very efficient real-time compression algorithms in the controller software, it can store a few days of continuous recording. In practice, the recording time is limited only by the battery capacity. Figure 6 presents the real view of the sensor with the SD card and cable. The sensor was enveloped in the transparent thermal shrink tube to provide electrical insulation, secure the cable connection, and protect the part of the micro-SD card sticking out.

Another useful feature of the flight controller is powering flexibility. It accepts the DC power voltage in the range of 8–25 V, equivalent to a 2S–6S Li-ion battery (six cells connected in serial). This makes it possible to power it from a variety of power sources. Simultaneously, the controller is able to monitor the battery voltage and cut of recording if the battery is depleted. Removes the need to use an additional battery management system.

This version is equipped with an STM32F405 microcontroller, a combined MPU6000 IMU device with MEMS accelerometers and gyroscopes. Optionally, it can be equipped with a magnetometer to improve the orientation on the horizontal plane.

Drone flight controllers must work in very dynamic environment with much higher dynamic than motion of the human body. For this reason, there is a requirement to use a high refresh rate for the sensors. To effectively use the fusion algorithms to detect 3D attitude, the refresh rate of 8 kHz for gyroscopes and 8 kHz for accelerometers are used. More recent versions of flight controllers use the 32 kHz gyroscope refresh rate. The microcontroller power is sufficient not only to calculate the attitude data in real time, but also to control the drone hardware, so for human motion capture only the small fraction of computational power is utilized, which allows to implement additional feature in the controller software in the future.

The attitude of the sensor can be described in output by the Euler angles or by quaternions. The more subtle 3D motion of human joints can be represented in the Motion Sphere, a novel trajectory-based visualization technique, to represent human motion in a unit sphere [18]. Quaternions are convenient if the measured angles are in a wide range, and gimbal lock can be an issue. For the purposes of this study, the Euler angles were selected because the range of measured angles is relatively low and the gimbal lock cannot occur. Moreover, the estimation of an angular errors in degrees is more informative.

The flight controller works under the control of the customized Betaflight software in version 4.2.8. All tasks related to controlling the drone motors were disabled to save power and not disturb the sampling process.

The hardware setup user for measurement is presented in Figure 7. It contains five sensors connected by cables to the battery and the remote control receiver. In the cable, 3 wires were used to connect the ground, positive voltage, and trigger signal.

To trigger the recording, the remote control receiver FS-RX2A that works with AFHDS 2A was used (Figure 8). The receiver works in PPM mode and is able to pass to the controller up to 8 channels of data through the single line, which allows implementation of not only triggering, but also various additional features such as, for example, remote reset of the sensor coordinate system.

The useful feature of the 3-wire connection between the central unit and the sensor is the possibility of using a daisy chain configuration (Figure 7). The communication between the receiver and sensors is unidirectional, so there is no need to use a separate cable for each sensor. This allows, for example, to use a single 3-wire cable to connect three sensors located on one leg.

The important feature of the sensors with local recording is the synchronization of the measurement. The flight controller has no precise real-time clock on the board, and the measurement of time is based only on the internal timer of the microcontroller. During long-time recording, the difference in internal clock frequencies results in desynchronization of the sensors.

This issue is solved by central triggering and stopping of the recording. even if the clock speeds are different, the moment in time of start and stop is synchronized by the remote control receiver signal. The flight controllers can be triggered and stopped many times, but each controller stores the subsequent recordings in numbered files.

After the completed recording session, the data can be downloaded from the sensor via the USB port or by removing the micro-SD cards and using the micro-SD card reader in the computer. The synchronization of the downloaded single recording is performed in the following steps:Checking of nominal recording time for each sensor *t_i_*;Calculation of average recording time $t_{av}$ for all sensors,
(1)$$t_{av}=\frac{\sum_{i=1}^{n}t_{i}}{n}$$
where  *n* is the number of sensors,  *i* is the index of sensor, *i* = 1…*n*,Adjusting the timescale $s_{i}$ of individual recordings use to modify the single sample time
(2)$$s_{i}=\frac{t_{av}}{t_{i}}$$
so all recordings have the same duration;Resampling all the recording to the frequency of 1000 Hz (1 ms sample time).

After this procedure, all samples from all recordings are synchronized. There is no guarantee that single sample time is exactly 1 millisecond, but time step accuracy is much better than precision of single microprocessor clock due to averaging of error. The stages of synchronization:Recording, different number of samples;Adjusting the frequency to the common average time;Resampling;
are presented in Figure 9.

This adjustment is made for each recording in the session separately based on the difference between the local start time and the local end time of the recording. The procedure is performed automatically by the script written in Python that downloads the data from the sensors. An example of the output of the synchronization procedure is presented in Figure 10.

The comparison of different sensors presented above with the proposed sensor is shown in Table 1. Wireless sensors, Xsens Awinda and Noraxon Ultium, are much more expensive solutions than the other ones listed in the table. Sensor sizes are comparable; however, internally powered sensors are bigger due to the size of internal battery.

The flight controller is more flat, only 8 mm thick, making it easier to install it on the body surface. It is also very lightweight; it decreases inertia influencing the acceleration measurement by sensors that are not rigidly connected to the body. Small inertia can improve the dynamic response of the sensors in cases where it is hard to fix the sensor to the body.

The proposed sensor based on the flight controller has also a unique ability to store measurement data internally for long-time measurements. One minute of recording requires 35 kB of storage. The micro-SD card with a capacity of 32 GB used in each sensor is sufficient to store 10 days of recording with a 1000 Hz sampling rate. In practice, the recording time is limited only by the power source. The measured power consumption of the single flight controller sensor during recording is 0.81 W. A set of 5 sensors, powered by a single central Li-Ion 3S battery 3600 mAh (mass 150 g), can record data for 9 h.

## 3. Measurements-Synchronization

In order to show the synchronization process, a simple measurement of sensors excited by acceleration pulses was carried out. The sensors were connected to each other, and the precise wired AHRS sensor, the Xsens MTi G-700, was attached to the FC sensors (Figure 11). The bundle of sensors was located on a flat, rigid surface and excited simultaneously by the series of nine impacts from the bottom of surface. The recorded signals before synchronization are shown in Figure 12. It can be seen that the last pulse is not perfectly synchronized for all sensors due to an inaccurate internal clock speed. The magnified first, middle, and last pulses are shown in Figure 13. The FC1 sensor (orange lines in Figure 13) shows significant increases with time desynchronization due to a lower internal clock speed.

The results of synchronization of flight controller sensors are presented in Figure 14. Additionally, the output of the reference MTi G-700 sensor equipped with an internal precise real-time clock was provided for comparison. In Figure 15 the magnified views of the first, middle, and last pulses are provided. The MTi recording was manually synchronized on the first pulse.

The FC recordings are synchronized within 1 ms, all the FC curves are overlapping in time. The increase in *t* between the reference MTi signal and the synchronized FC signals is a consequence of the inaccurate clock of the FC sensors. The synchronization error was in this case equal to 0.1%.

In cases where FC sensors are used to estimate the movement of the human body, synchronization between sensors is sufficient, and an error in the real duration of samples can be neglected. Otherwise, if additional external sensors must be synchronized with the FC recordings, the additional external synchronization technique must be adopted. The pre- and post-synchronization recordings were attached as Appendix A.

## 4. Measurements—Angular Accuracy

The angular accuracy was estimated in a similar way by comparison of FC measurement with the precise Xsens MTi G-700 wired sensor. The sensors were connected to each other and connected to the palm.

The series of dynamic movements of were performed by rotation of the forearm from the horizontal position to the approximately vertical position. The results of the angle of rotation in the vertical plane (pitch angle) obtained from the FC sensor and the reference MTi sensor are shown in Figure 16.

There was no external synchronization between the sensors during the measurement, and both signals were measured independently. The signals measured by the reference sensor and the FC sensor were manually synchronized, and the starting points were determined by minimizing the RMS of difference. The timescale of the FC signal was not corrected.

The difference between the angles provided by both sensors is presented in Figure 17. The standard deviation of the FC sensor from the reference sensor was equal to 1.9 deg (red dashed lines).

Outliers can be observed in the error estimation curve, and the maximum error values reach 9°. Outliers are not actually the result of an incorrect value of the estimated angle, but desynchronization artifacts. As mentioned earlier, the flight controller sensor does not have a precise real-time clock, so even with a short measurement, the reference sensor and the FC sensor are out of sync and the signals are locally shifted in time. For dynamic movements, even a small phase shift makes a significant difference in the measured angles.

This issue is presented in Figure 18, magnification of the signals and error from Figure 16 and Figure 17 at the moment of maximum error. It can be seen that the actual amplitude error is less than 3°, while the 9° error in the angle estimation is just a desynchronization effect. The out-of-sync error does not occur if all angles are measured simultaneously by a set of synchronized FC sensors.

To improve signal synchronization between the reference sensor and the FC sensor, the time scale of the FC signal can be corrected. In the case of a set of multiple FC sensors, the clock error is lowered by averaging the clock errors between the FC sensors.

Another test was carried out with the use of three sensors: flight controller sensor, commercial wireless Xsens MTw Awinda sensor, and reference Xsens MTi G-700 sensor with the use of the same setup: dynamic rotation of the forearm in vertical plane. The bundle of sensors is shown in Figure 19. The results of the measurement are presented in Figure 20.

The magnification of the single movement is shown in Figure 21. It can be observed that the deviations of the FC sensor signal and the commercial Awinda sensor from the reference MTi G-700 signal are comparable and equal to approximately 2°. The synchronized angle measurements were attached as Appendix A.

It should be highlighted that the flight controller sensor used as a set of sensors is very well-synchronized by the common electrical signal at the start and the end of recording, but if specific research requires synchronization of this measurement with external signals such as force measurement, optical systems, or electromyography (EMG), it became problematic because of inaccurate time measurement of each sensor.

The drift of integrated quantities can be potentially affected by the inaccuracy of the internal clock. However, the inaccuracy of the clock is on the level of 0.1 %, so the accumulated error should increase quite slowly. The drift of roll and pitch is limited by gravity vector detection, but the drift of yaw (azimuth) cannot be corrected and it can cause increasing difference between sensors. The influence of desynchronization on the attitude estimation should be investigated in further studies.

## 5. Measurements—Application Example

To test the sensor, the experiment was set up to measure the accelerations, positions, and orientations of the human body during motion. The measurements presented below are intended to highlight the usability of FC sensors and discuss the technical issues of their use. The tests were carried out with a single person. The full biomechanical research will be published in a separate paper.

The tests were carried out during normal gait on a flat surface. Five sensors were used, located on the head, neck, lumbar part of the spine, and feet. The sensor location is presented in Figure 22.

Sensors in the feet, tailbone, and neck were connected to the skin using double-sided self-adhesive tape. The sensor located on the top of the head was fixed to the head using a tight elastic hat. It should be mentioned that this method of sensor fixation does not provide a rigid connection to the bones because, in the case of a highly dynamic load, the inertial forces cause relative movement between the skin with the fixed sensor and the bone. It works in that case as a mechanical damper and reduces the measured acceleration.

The tests were carried out during natural gait. The tested person made four passages, 40 steps each on a hard surface with bare feet and in sport shoes. The three components of the acceleration were measured in the sensor coordinate system and in the Earth coordinate system, this was possible because the sensor to estimate real-time attitude in the Earth coordinate system. The impact of the feet on the ground initiated the strain wave that travels through the body from the feet toward the head. In the tests, the vertical acceleration at different locations in the body and the transmission of the strain wave during the human body were investigated. Note that without knowledge of the real-time orientation of each sensor, it would be impossible to separate the only vertical component of the acceleration.

Example results of angle measurements are presented in Figure 23 and Figure 24. Measurements were taken during the same passage. The temporal resolution of 1 kHz is, in this case, higher than necessary. Sensors with a temporal resolution of 100 Hz can provide sufficient reproduction of the walk angles.

Much more interesting are the results of acceleration measurement. It should be highlighted that measurement of 3D acceleration, in addition to the angle estimation, can provide very useful information about human motion. To show an example of this information, the data for the passage in sport shoes and with bare feet are presented below.

Although pose estimation is relatively slow-changing, the accelerations even during the walk can contain very short peaks. To detect the correct maximum values of these peaks, a much higher temporal resolution is required. In Figure 25 and Figure 26 the vertical accelerations of each sensor are presented during the passage with bare feet and sports shoes.

Measurements were stable with good repetition between steps. The left foot provides slightly lower top values, which can be evidence of nonsymmetrical gait. Much higher values of top acceleration are visible in the passage with base feet, which is a result of impact of the heels in the hard surface. In the case of bare feet, the maximum acceleration value in the foot reached 95 m/s^2^, about 70% higher than in the case of sports shoes, when it reached only 55 m/s^2^.

To visualize the acceleration in the head, neck, and tailbone, it was shown in Figure 27 and Figure 28 without dominating foot signals.

In both cases, the highest vertical acceleration was detected at the top of the head. In the same way as for feet, the accelerations in the case of bare feet are about 60% higher than acceleration in sport shoes.

Repeatability of the acceleration measurements was analyzed by shifting the vertical accelerations measured on the head during each step to each other (Figure 29). The peak acceleration during each step was marked as zero time and all the steps of the passage were synchronized by the peak times.

It can be observed that the shape of the peak acceleration is exactly the same, only the amplitude is different. Maximum values are in the range of 18.5 to 23 g. The total time of the vertical impact event that acts on the head is equal to 50 milliseconds. The temporal resolution of 1000 Hz is sufficient to detect the actual peak value and precisely resemble the shape of the acceleration curve.

The numerical values of the maximum accelerations acquired during the passage with bare feet and sport shoes are presented in Table 2 and Table 3. In Table 4 the average values for 10 steps are calculated and the attenuation ratios between the sport shoes case and the bare feet case are presented.

The results of the acceleration measurement are presented graphically in Figure 30 and Figure 31. Maximum vertical acceleration levels were shown at all levels of the body.

The interesting fact is that the accelerations are not monotonically lowered/damped from feet to head. To visualize the attenuation of vertical acceleration between different levels, the ratios for various pairs were calculated and presented in Table 5.

The ratios above 100% in the case of pairs neck/tailbone and head/tailbone means that the acceleration measure on the tailbone probably is lower than the real one. The reason is probably the connection of the sensor to the skin and the relative movement between the skin and tailbone. During impact, the sensor can move with the skin in a vertical direction, and the measured acceleration is lower than the acceleration of the bone.

## 6. Discussion

The purpose of the measurements presented in Section 3 was to test and present the ability of the customized flight controller to work as a human body motion sensor. In addition to the most attractive feature, low cost, it is able to provide temporal high-resolution data in two domains:Attitude of the sensors, which in connection with specialized software can be used to resemble the kinematics of all parts of the human body, for example, for biomechanical research purposes; the standard deviation of dynamic angle estimation at the level of 1.9 deg is sufficient for this purpose;Acceleration at the sensors, which can be used to estimate the actual forces acting on the joints during movement and provide data for dynamical simulations.

It should be stated that not all biomechanical attitude sensors are able to also provide accelerations at the high rate. For example, the set of 17 Xsens Awinda sensors provide accelerations at rate of 60 Hz.

As presented in Table 1, the proposed sensor is capable of providing accelerations and attitude at a higher rate than much more expensive sensors. As presented in the simple walking study above, the acceleration signal sampled at the rate of 1000 Hz is able to detect the peak acceleration during the impact of the bare foot into the hard surface; however, the slower sampling fails at this task. The sensor response during the single impact of the naked foot on a hard surface is shown in Figure 32.

The impact event is very short in time, taking about 40 ms, so its shape is described by 40 samples. It reveals the necessity of usage of high temporal resolution during analysis of accelerations in human body motion. In the presented case, the 1 kHz signal acquisition frequency is sufficient and capable of properly capturing the maximum acceleration value.

The question can be asked if, in application for much more dynamic events, such as the impact of the rugby players, the higher rates of application can be required. The answer is that the higher rates with the sensors similar to those presented are probably useless because of the sensors have a mass of about 10 g. The inertia of the sensor acts as a mechanical low-pass filter, and even if the higher frequency accelerations appear on the human body, transfer of this acceleration is limited by relative movement of the sensor and the skin. In order to measure higher frequencies of acceleration, much lighter sensors or much more rigid connection of the sensor to the body, or even to the skeleton, is required. This effect is visible in the acceleration analysis of the neck and head.

The acceleration wave traveling from the foot impact through the body should theoretically be damped as it travels upward to the head, while the accelerations measured on the head were higher than the accelerations measured on the neck.

The neck sensor was connected to the skin in a vertical position, so for vertical acceleration, the relative movement of the neck and the sensor was possible because of the elasticity of the skin. As a result, the vertical accelerations measured in the neck were probably underestimated.

On the head, the sensor was installed in a horizontal position at the top of the skull. When the acceleration wave reached the skull, the relative movement of the sensor and the skin in the direction normal to the skin was limited much more than the relative movement of the skin on the neck and the neck sensor in the direction tangent to the skin. This may explain why the higher acceleration values measured at the location were more distant from the impact point.

## 7. Conclusions

The presented application of the adapted flight controller to the function of the human body movement sensor proves that the sensor is sufficient for this role. While the temporal resolution of 1 kHz for human body dynamic pose estimation is higher than required, in the case of accelerations the situation is different. Even during undynamic movement of the body during walking on a flat surface, 1 kHz temporal resolution is required to capture foot impact event correctly. The interesting conclusion is that usage of lower frequency sensors, with the acquisition frequency on the level of 100 Hz can result in the significant underestimation of the measured acceleration. The measured standard deviation of the angle estimation error was equal to 1.9°.

The presented configuration of the sensors with local recording and centralized triggering provides simultaneously long-time recording, limited in practice only by the capacity of the battery, and automatic synchronization of all the recorded signals. The number of sensors is unlimited because there is no bottleneck in the form of bandwidth, as in the case of wireless sensors or a central recording unit. It seems to be an optimal solution, especially in the case of long-time and high temporal resolution recording. The disadvantage of this setup is, of course, a necessity of the cable connection for central powering and triggering of the system, but it is the price for a relatively small sensor footprint and low mass. Cables are not used for high-speed data transfer, so they can be relatively thin and lightweight.

The internal synchronization of the flight controller sensors is very good because it is triggered and stopped by a common electrical signal. However, synchronization with external sensors is difficult because there is no precise real-time clock on board. The error in the measurement of the average time by the sensors was equal to 0.1%.

The ability of the sensors to provide not only acceleration, but also stable attitude and acceleration in the Earth coordinate system is much better than standard 3 axis accelerometers. In attitude sensors, the most problematic is the estimation of the azimuth (orientation in horizontal plane), but it can be partially solved using the flight controller equipped with a magnetometer.

The relationship of cost/effect at the cost of USD 50 per single smart sensor with a long-term recorder is excellent. The usage of this type of sensor is possible not only in the human body movement, but also in other areas, such as monitoring robot or vehicle motion. The ability of relatively long recording time (up to 10 days) without external recorder and operation on a central battery of practically unlimited capacity makes it a very versatile device.

The relevance of this work is not in the results of human body measurement, it is provided only as an example of the use of an FC sensor. More extensive human body measurement research, focused on biomechanical aspects, will be carried out with the use of FC sensors and published in separate papers. Here, only the technical aspects of the flight controller sensor are discussed, and this is the main purpose of this paper. This can be especially useful for researchers with a limited budget.

## Figures and Tables

**Figure 1 sensors-23-02342-f001:**
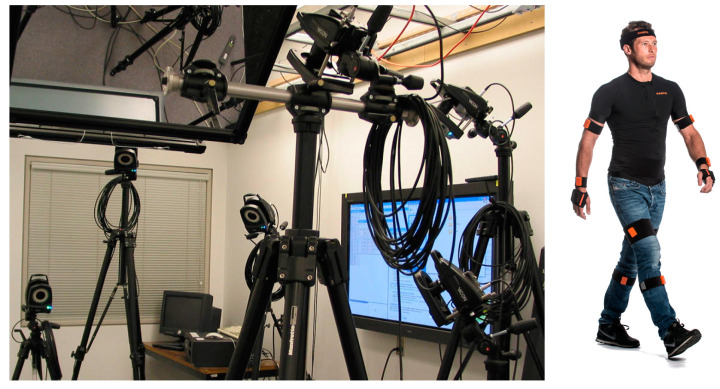
The set of sensors from the optical motion tracking system: VICON (**left**) and IMU-based sensors mounted on the human body (**right**).

**Figure 2 sensors-23-02342-f002:**
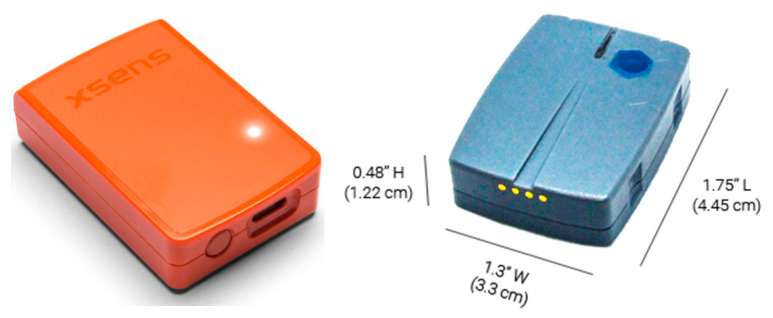
MTw Awinda sensor 47 × 30 × 13 mm, mass 16 g (**left**); and Noraxon Ultium Motion sensor 44.5 × 33 × 12.2 mm, mass 19g (**right**).

**Figure 3 sensors-23-02342-f003:**
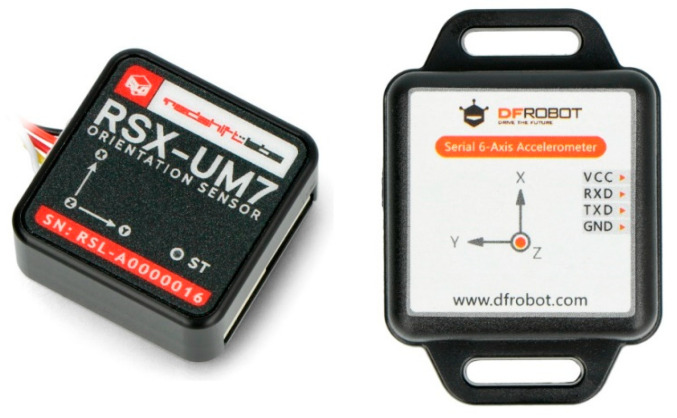
RedShift Labs UM7, 28 × 28 × 11 mm, mass 7.5 g (**left**); and DFRobot SEN0386 sensor, 51.3 × 36 × 10 mm, mass 18g (**right**).

**Figure 4 sensors-23-02342-f004:**
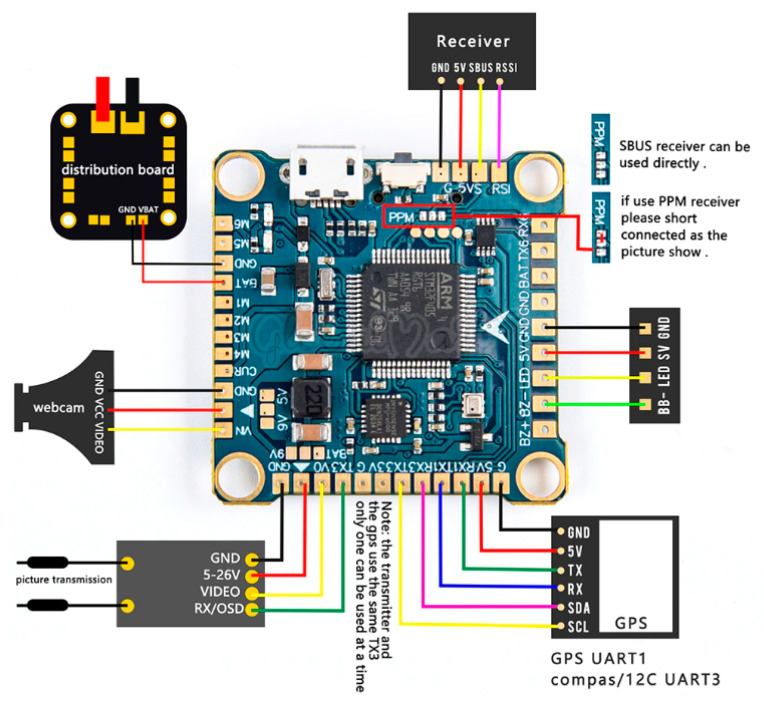
Layout and connectivity ports of the flight controller.

**Figure 5 sensors-23-02342-f005:**
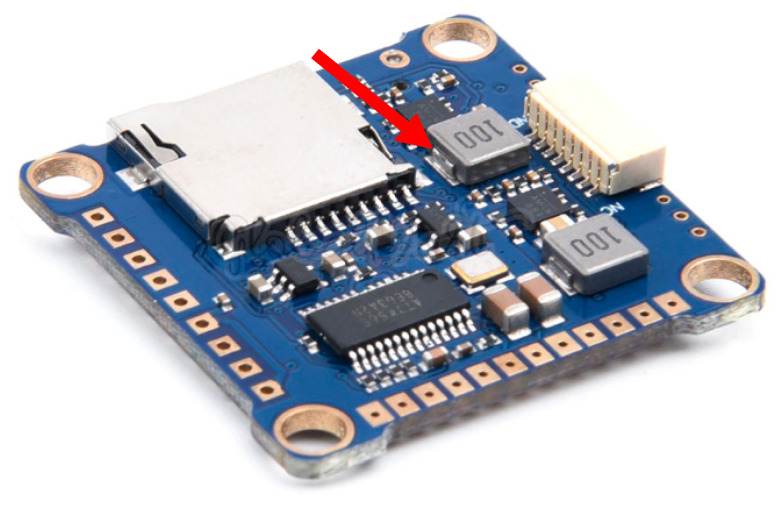
The SD card slot on the bottom side of the flight controller.

**Figure 6 sensors-23-02342-f006:**
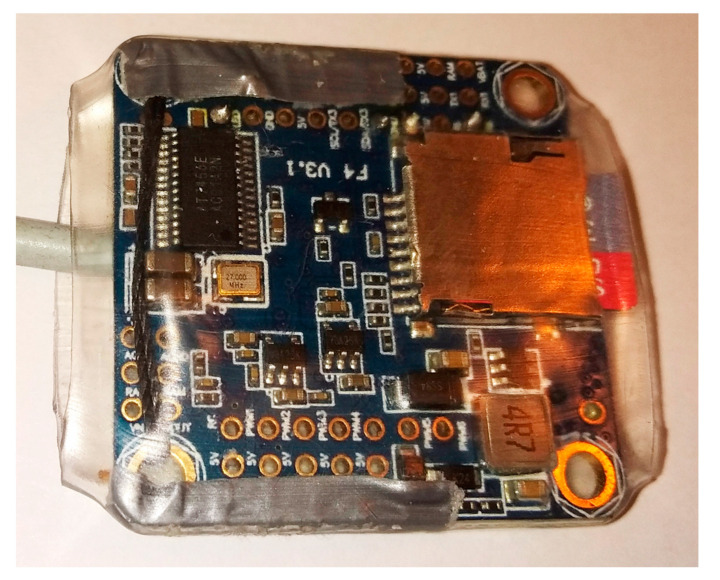
View of the cabled sensor secured by a transparent shrink tube, with a 32 GB micro-SD card in the slot for local recording.

**Figure 7 sensors-23-02342-f007:**
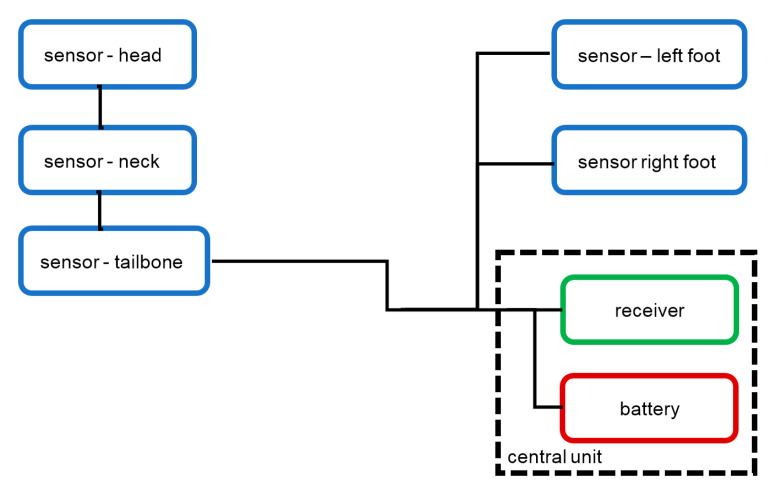
The layout of the measurement system.

**Figure 8 sensors-23-02342-f008:**
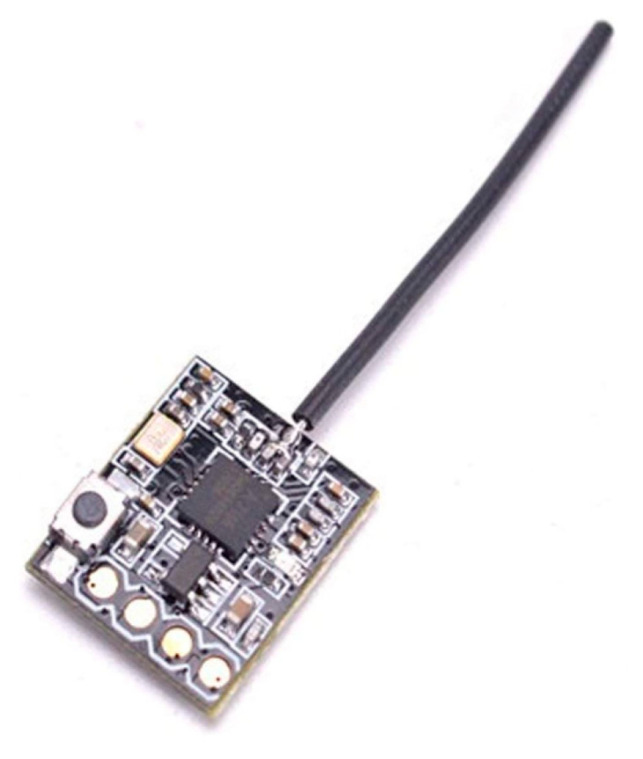
Remote control receiver used to trigger the sensor system.

**Figure 9 sensors-23-02342-f009:**
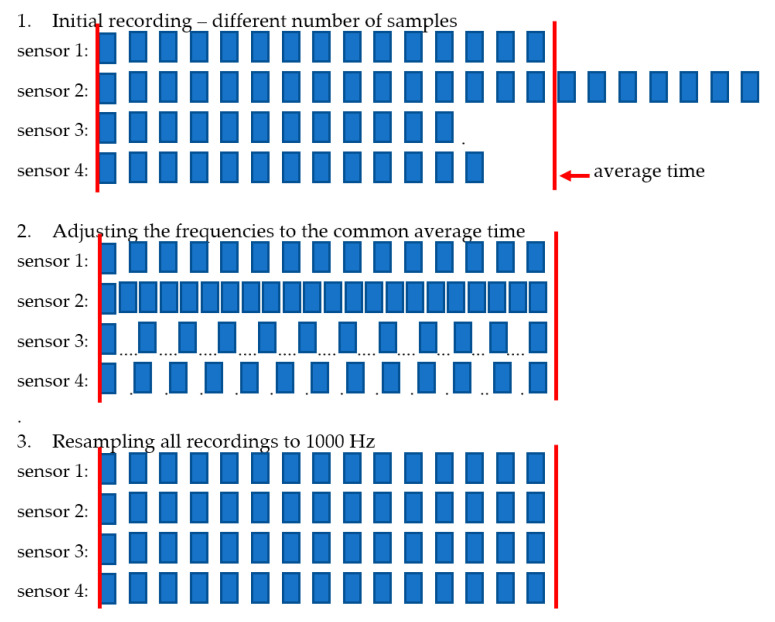
The stages of recording synchronization.

**Figure 10 sensors-23-02342-f010:**
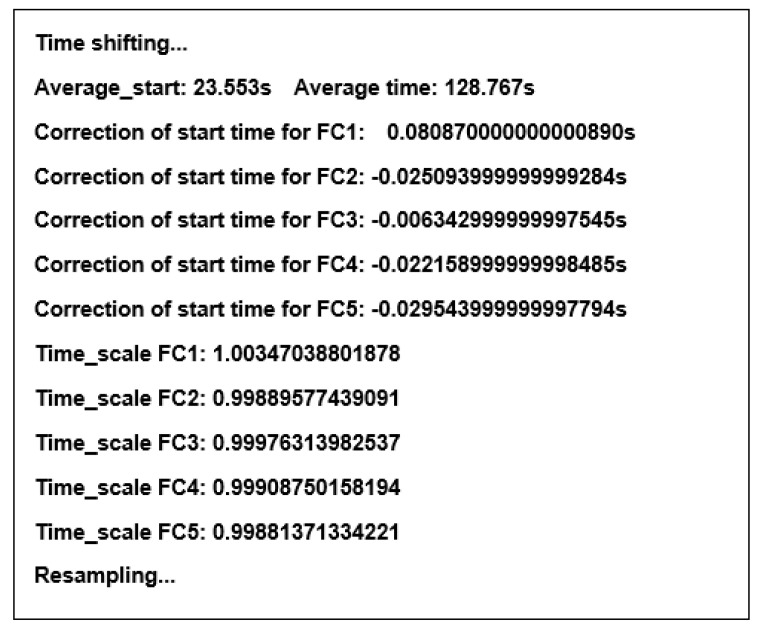
The example output of the synchronization procedure in Python.

**Figure 11 sensors-23-02342-f011:**
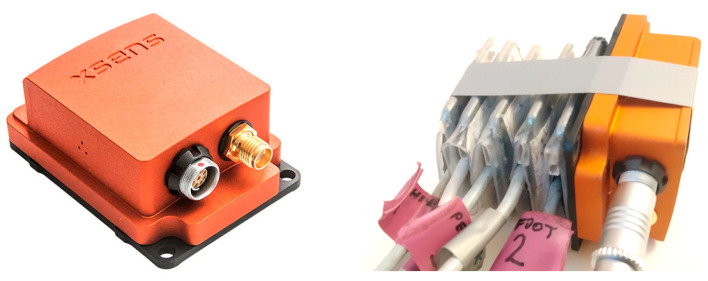
The reference Xsens MTi G-700 sensor (**left**); and the flight controller sensors for synchronization measurement (**right**).

**Figure 12 sensors-23-02342-f012:**
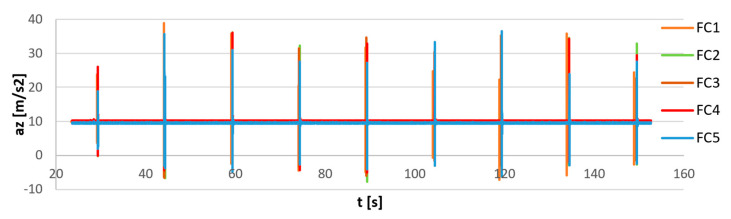
Raw measurement of impulse excitation (acceleration on the Z axis) of 5 flight controller sensors (FC1–FC5) before synchronization.

**Figure 13 sensors-23-02342-f013:**
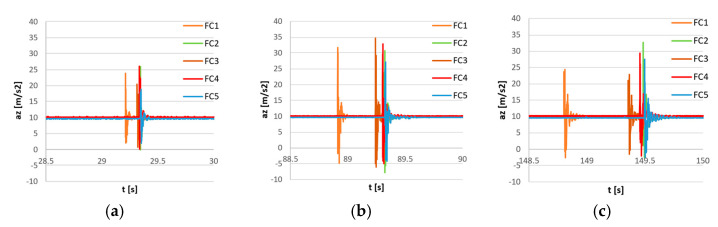
The raw measurement of impulse excitation (acceleration in Z-axis) of 5 flight controller sensors (FC1–FC5) before synchronization. Magnifications of the first pulse (**a**), the middle pulse (**b**), and the last pulse (**c**).

**Figure 14 sensors-23-02342-f014:**
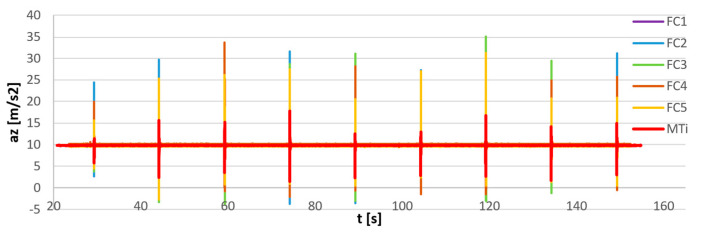
Measurement of impulse excitation of five flight controller sensors (FC1–FC5) after synchronization and reference signal measured with the Xsens MTi-G700 sensor (MTi).

**Figure 15 sensors-23-02342-f015:**
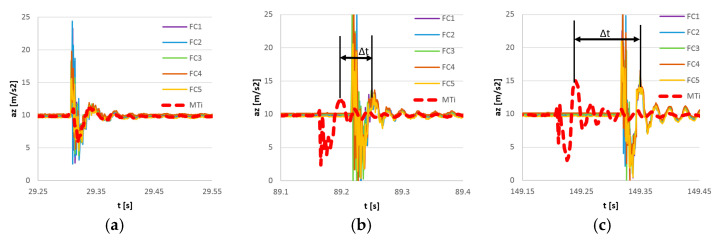
Measurement of impulse excitation by five flight controller sensors (FC1–FC5) after the synchronization and reference signal measured with the Xsens MTi-G700 sensor (MTi). Magnification of the initial (**a**), middle (**b**), and last pulse (**c**). ∆t: shift between the reference pulse and the FC pulse.

**Figure 16 sensors-23-02342-f016:**
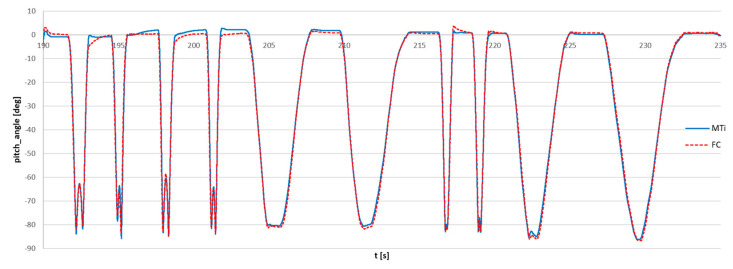
Comparison of dynamic angle measurement using the flight controller sensor (FC) and reference Xsens MTi G-700 sensor (MTi).

**Figure 17 sensors-23-02342-f017:**
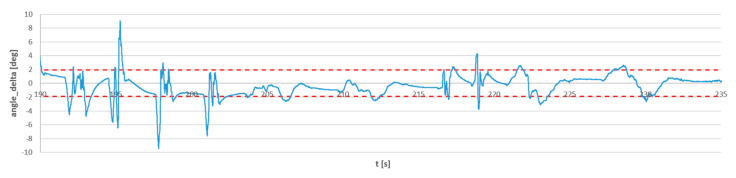
The angle deviation between the FC sensor and the reference MTi G-700 sensor. The red lines represent the standard deviation of the FC signal according to the reference MTi signal.

**Figure 18 sensors-23-02342-f018:**
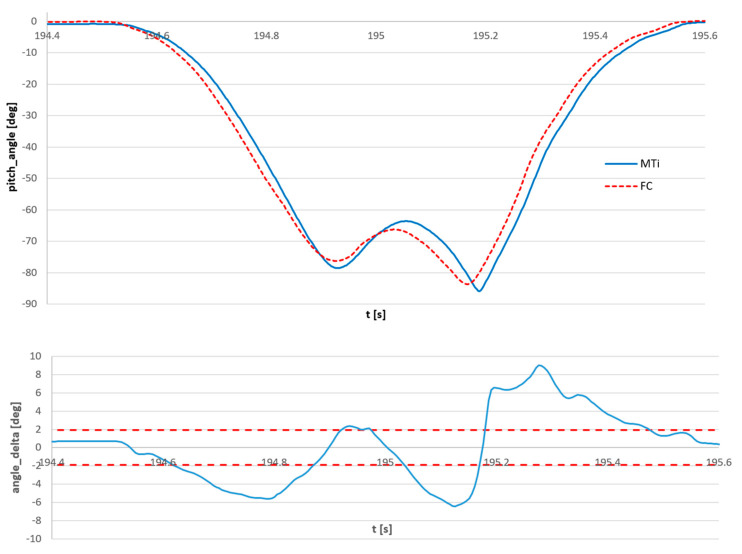
The magnified view at the time of maximum error from Figure 17, the signals measured by the reference sensor and the FC sensor (**top**) and error of estimation (**bottom**). The red dashed lines in the bottom view represent the standard deviation of the FC signal according to the reference MTi signal.

**Figure 19 sensors-23-02342-f019:**
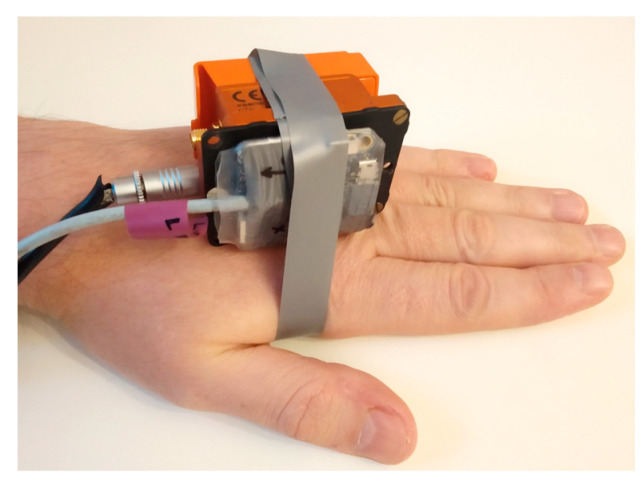
The MTi G-700 sensor and the set of sensors connected to the palm: MTi-G700 sensor, Xsens Awinda wireless sensor, and flight controller sensor.

**Figure 20 sensors-23-02342-f020:**
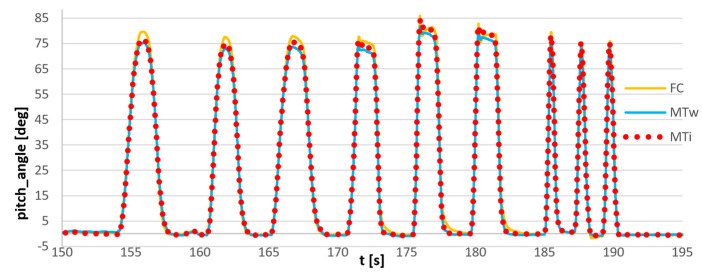
Synchronized measurement of pitch angle: MTi: reference signal, Xsens MTi-G700 sensor; MTw: Xsens Awinda wireless sensor; FC: flight controller sensor.

**Figure 21 sensors-23-02342-f021:**
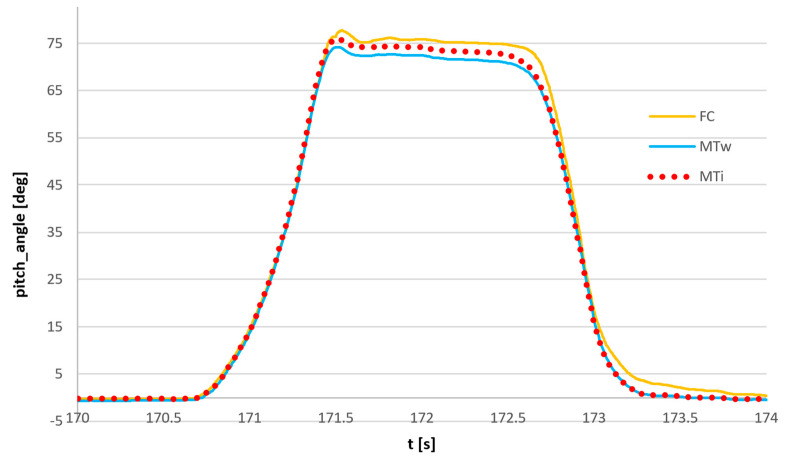
Synchronized measurement of the pitch angle, magnification of the single movement: MTi: reference signal, Xsens MTi-G700 sensor; MTw: Xsens Awinda wireless sensor; FC: flight controller sensor.

**Figure 22 sensors-23-02342-f022:**
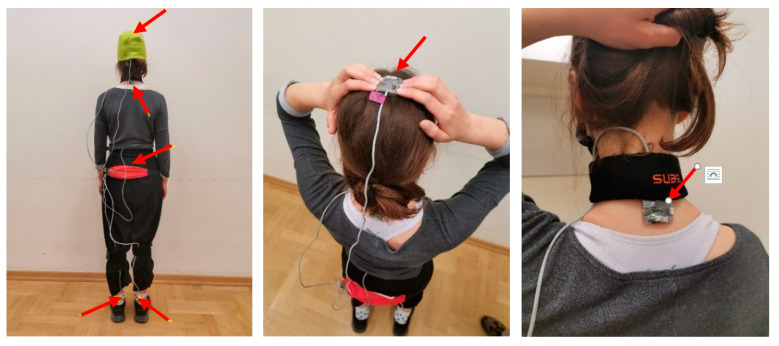
Location of sensors on the human body (**left**), location of the sensor on the head (**middle**) and on the neck (**right**).

**Figure 23 sensors-23-02342-f023:**
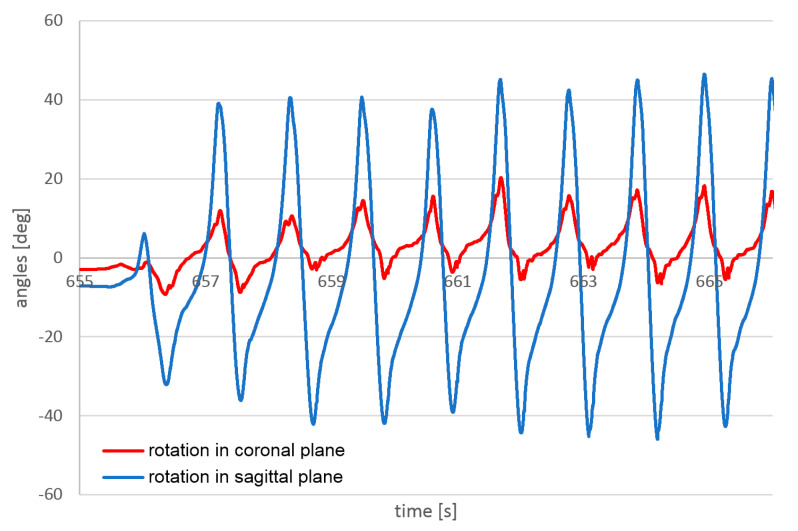
Angles measured on the left foot during walking by the flight controller.

**Figure 24 sensors-23-02342-f024:**
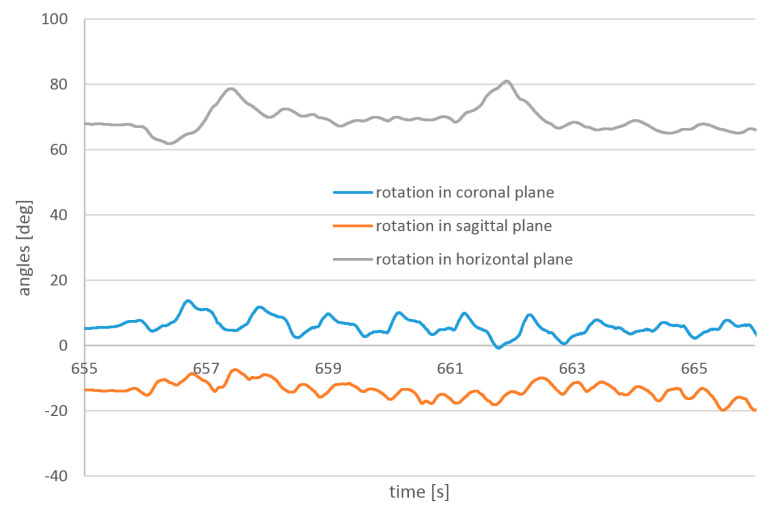
Angles measured on the head during walking by the flight controller.

**Figure 25 sensors-23-02342-f025:**
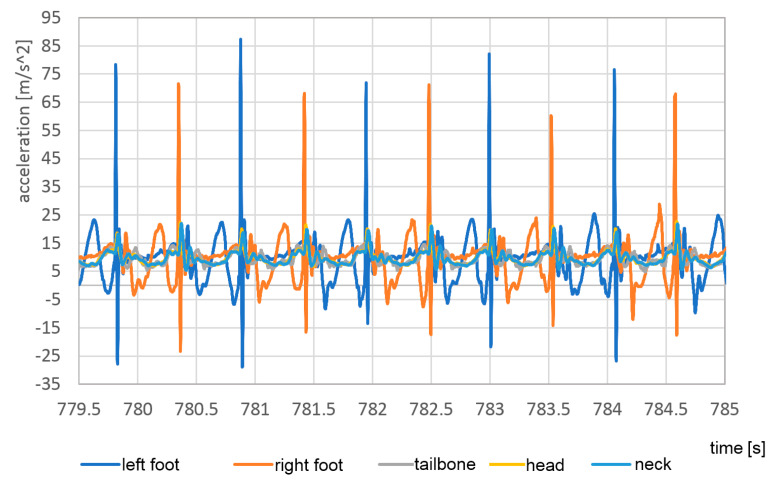
Vertical component of acceleration of each sensor during the passage with bare feet.

**Figure 26 sensors-23-02342-f026:**
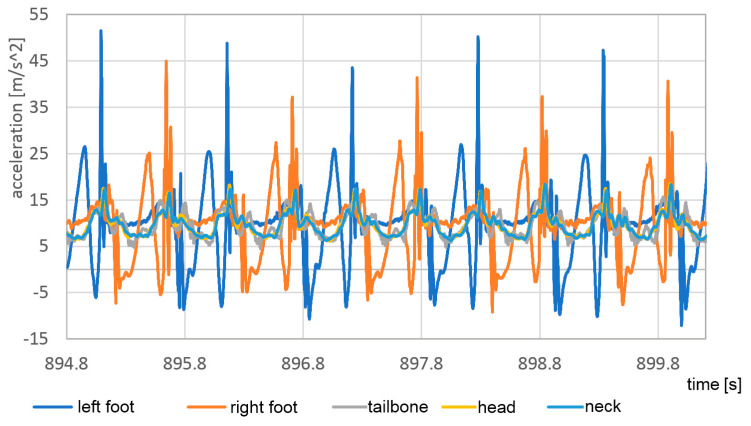
Vertical component of acceleration of each sensor during the passage in sport shoes.

**Figure 27 sensors-23-02342-f027:**
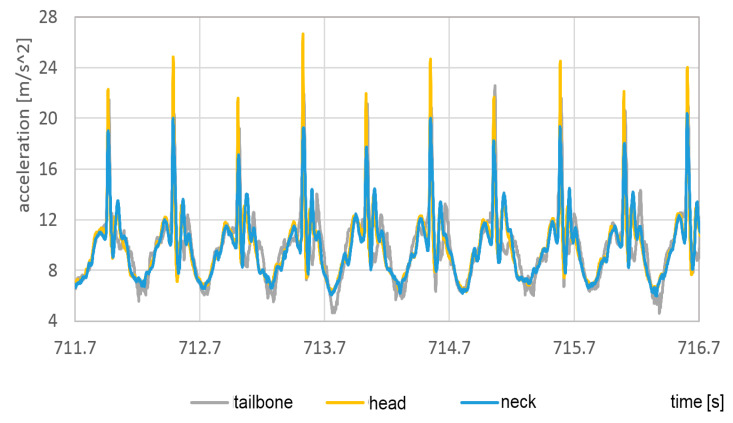
Vertical component of the head, neck, and tailbone during passage with bare feet.

**Figure 28 sensors-23-02342-f028:**
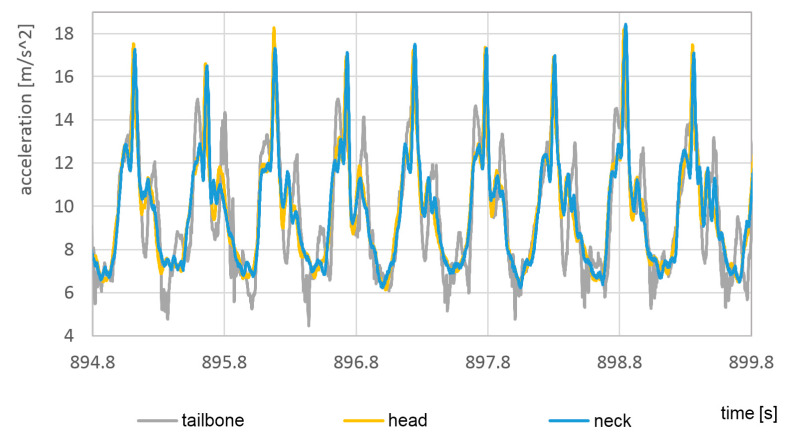
The vertical component of the acceleration of head, neck, and tailbone during the passage in sports shoes.

**Figure 29 sensors-23-02342-f029:**
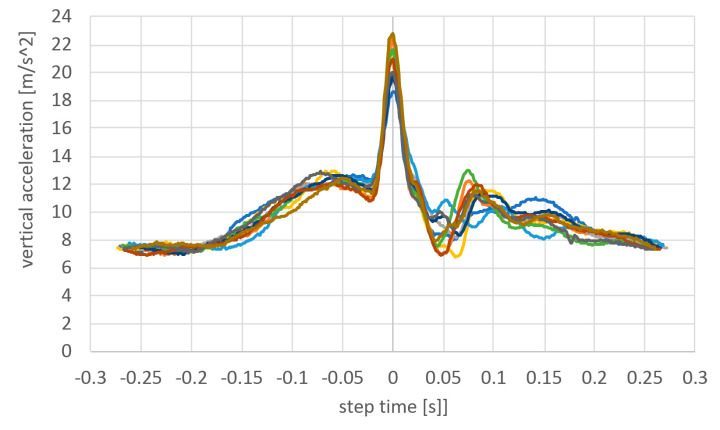
The vertical component of the head acceleration during 10 subsequent steps of the single passage.

**Figure 30 sensors-23-02342-f030:**
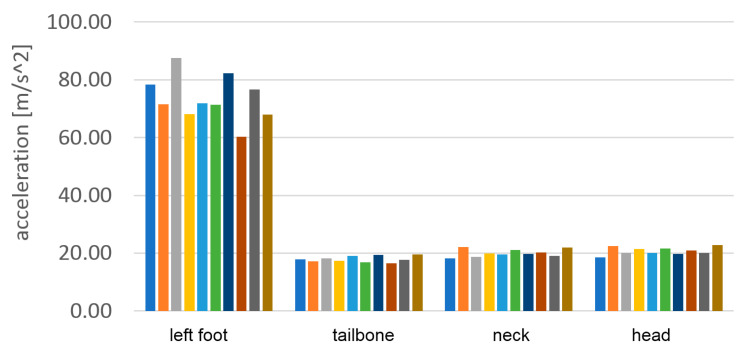
The vertical component of the head acceleration during 10 subsequent steps of the single passage with bare feet.

**Figure 31 sensors-23-02342-f031:**
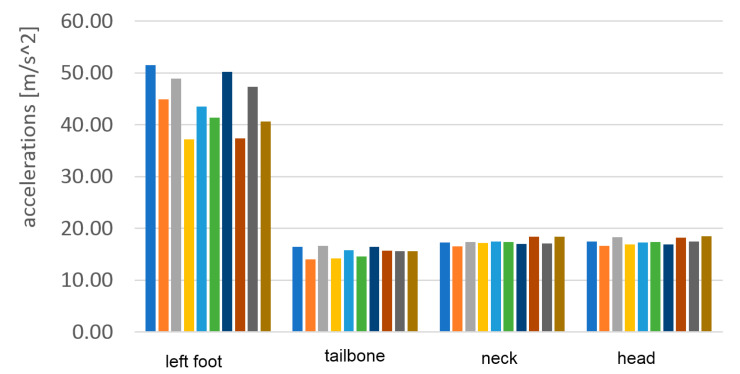
The vertical component of the head acceleration during 10 subsequent steps of the single passage in sports shoes.

**Figure 32 sensors-23-02342-f032:**
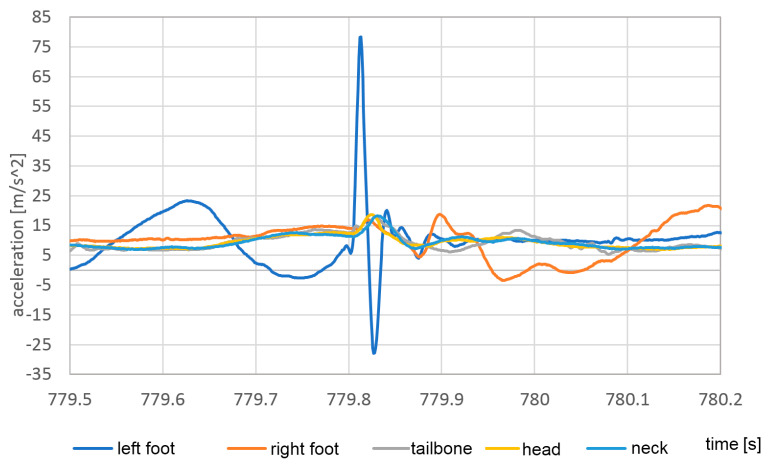
The vertical response of the sensors during the single impact of the bare right foot onto a hard surface.

**Table 1 sensors-23-02342-t001:** Comparison of selected movement sensors.

Sensor	Size [mm]	Mass [g]	Local Battery	Local Recording	Wireless	Frequency [Hz]
Awinda	47 × 30 × 13	16	yes	no	yes	120
Ultium	44.5 × 33 × 12.2	19	yes	no	yes	100
UM7	28 × 28 × 11	7.5	no	no	no	500
SEN0386	51.3 × 36 × 10	18	no	no	no	200
proposed	38 × 38 × 8	**8**	no	**yes**	no	**1000**

The advantages of proposed sensor are highlighted in bold.

**Table 2 sensors-23-02342-t002:** Top acceleration: bare feet [m/s^2^].

Step	1	2	3	4	5	6	7	8	9	10
feet	78.39	71.56	87.47	68.14	71.87	71.30	82.19	60.32	76.66	68.00
tailbone	17.88	17.21	18.27	17.38	19.00	16.85	19.41	16.53	17.76	19.63
neck	18.21	22.12	18.73	19.95	19.61	21.16	19.81	20.25	19.02	21.94
head	18.61	22.41	20.02	21.49	20.00	21.63	19.74	20.99	20.02	22.78

**Table 3 sensors-23-02342-t003:** Top acceleration: sport shoes [m/s^2^].

Step	1	2	3	4	5	6	7	8	9	10
feet	51.50	44.94	48.88	37.23	43.53	41.36	50.26	37.34	47.38	40.67
tailbone	16.39	13.97	16.62	14.24	15.81	14.62	16.40	15.69	15.56	15.63
neck	17.29	16.51	17.33	17.14	17.49	17.34	16.99	18.43	17.11	18.39
head	17.50	16.61	18.25	16.93	17.25	17.33	16.88	18.24	17.47	18.53

**Table 4 sensors-23-02342-t004:** Ratios of the top acceleration damping between bare feet and sport shoes.

	Bare Feet		Sport Shoes		Ratio [%]
	Average [m/s^2^]	Std Deviation [m/s^2^]	Average [m/s^2^]	Std Deviation [m/s^2^]	
step	73.59	7.81	44.31	5.15	60.21
feet	17.99	1.07	15.49	0.93	86.12
tailbone	20.08	1.31	17.40	0.59	86.66
neck	20.77	1.31	17.50	0.65	84.26

**Table 5 sensors-23-02342-t005:** Acceleration ratios measured by higher sensor to lower sensor: various pairs [%].

	Bare Feet	Sport Shoes
tailbone/feet	24.45	34.97
neck/feet	27.29	39.27
head/feet	28.22	39.50
neck/tailbone	111.61	112.31
head/tailbone	115.45	112.95
head/neck	103.44	100.57

In color the pairs with significantly higher acceleration on the upper sensor.

## Data Availability

Not applicable.

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
