# Peer review of "Flight Controller as a Low-Cost IMU Sensor for Human Motion Measurement"

_sensors, 2023, doi:10.3390/s23042342_

Round 1
Reviewer 1 Report (Previous Reviewer 2)
I feel that the manuscript is improved to a great extent compared to its earlier version. The authors have done a commendable work
Author Response
Dear Reviever,
Thank you for your positive responce.
Best regards
Reviewer 2 Report (New Reviewer)
Dear author,
Thank for your work on a flight controller for use for human motion tracking. I do have some feedback for your work:
- The introduction does not highlight which research gap your work helps to fill.
- Throughout the manuscript you use different sensors to compare to (Xsens Awinda (not Avinda), MTi G-700, Noraxon Ultium Motion), however, it is unclear why different tests require different reference IMUs.
- A lot of figures provide very little detail, for example 12/13, where it makes it hard to interpretate the meaning of those.
- Various figures to not have a legend, while multiple bars/lines are presented, which makes it very hard to interpret.
- There are also figures of acceleration between sport shoes/bare feet, where does this fit the purpose of the work?
In general, your work would benefit from restructuring and improving the description of relevance of this work.
Author Response
Reply is in the attached file.

Reviewer 3 Report (New Reviewer)
The author presented a novel work on the adoption of flight controllers applied to the human body serving as tracking systems. The results show that the proposed system (with synchronization software) is valid for the application and a comparison with a reference system was made to validate the results.
- Check again the whole document for typos and English rewriting of some sentences. I spotted some from page 1 to 14, but fewer later. I suggest Grammarly or other grammar-check software.
- I suggest presenting clearer the purpose of the paper at the end of the introduction, maybe also referring to what flight controller attitude is and how it is related to standard IMU measurements.
- It is not clear how many subjects participated in the study, what the experiment concerned (i.e. moving from point A to B following a straight line), and the parameters of the tests. Add another subparagraph at the start of section 5 with such details explained fully.
- The graphs provided in section 5 seem to refer only to one test of a particular subject, so they're useful for understanding but are not statistically interesting. It is crucial to estimate the variation of the proposed system from the reference across subjects and across tests (usually at least 30 repetitions of the same test should be conducted by the same subject; however this may be unpractical so I suggest at least 10 repetitions of the same test per subject). I also recommend at least 10 subjects equally divided by gender. Afterward, I recommend performing an ANOVA test and discussing the results (they can be tables, albeit graphs are easier to understand). For example, figures 30 and 31 refer to 10 consecutive steps of the same test of the same subject; hence, they're not statistically valid. I recommend doing the same for each subject, extracting the mean and deviation of the acceleration and plotting the results as candle plots/box plots.
Round 2
Reviewer 2 Report (New Reviewer)
Dear authors,
Thank you for taking my feedback into account for the revision of the manuscript. A lot of questions have been clarified in the latest version of the manuscript. I do have some minor comments on the manuscript:
- XSENS/XSENX is used as spelling throughout the manuscript, which should be written as Xsens.
- line 46 on page 2 says gyros, where you say gyroscopes throughout the manuscript.
- line 414-417 on page 15 states that MTw awinda does not measure acceleration, which is incorrect as this can be obtained from those sensors as well.
- It is clearly stated that inaccurate internal clocks affect the soundness of obtained duration of samples. Which should be highlighted better in the discussion as this does not hugely affects the acceleration but has increasing impacts on integrated quantities, e.g., velocity, position and orientation. These are quantities of interest in human motion and as stated in line 378/379 on page 13 the measurement setup was chosen such that those integrated quantities are analyzed, while the results do not show position outcomes. Therefore, the discussion should comment on this limitation.
Thank you for considering the feedback.
Author Response
Dear Reviewer #2,
Thank you for your comments.
- XSENS/XSENX is used as spelling throughout the manuscript, which should be written as Xsens.
Spelling has been corrected to Xsens.
- line 46 on page 2 says gyros, where you say gyroscopes throughout the manuscript.
Corrected: gyros-> gyroscopes
- line 414-417 on page 15 states that MTw awinda does not measure acceleration, which is incorrect as this can be obtained from those sensors as well.
This point is interesting. My version of MTw Awinda is around 10 years old. I know that Avinda description says that acceleration can be obtained (however not a pure accelerations but obtained from SDI).
I’m pretty sure that my original version of Awinda was not providing the acceleration at all. In the configuration software the options for acceleration are greyed out (not active) and there is no way to activate it. The MTi G-700, in the same software, provides accelerations without any problems.
I’ve checked it on the newest version of MT manager, and it works (Awinda accelerations) so it was a software issue.
However, even if Awinda is able to provide acceleration, the rate of acquisition is too low to make this accelerations useful for dynamic events, it was pointed out in the paper.
In the paper the correction was introduced, information that Awinda is not providing the accelerations was removed.
- It is clearly stated that inaccurate internal clocks affect the soundness of obtained duration of samples. Which should be highlighted better in the discussion as this does not hugely affects the acceleration but has increasing impacts on integrated quantities, e.g., velocity, position and orientation. These are quantities of interest in human motion and as stated in line 378/379 on page 13 the measurement setup was chosen such that those integrated quantities are analyzed, while the results do not show position outcomes. Therefore, the discussion should comment on this limitation.
Good point. The drift of integrated quantities can be potentially affected by inaccuracy of internal clock. However, the inaccuracy of the clock is on the level of 0.1 %, so the accumulated error should increase quite slowly. The drift of roll and pitch is limited by gravity vector detection, but the drift of yaw (azimuth) cannot be corrected and it can cause increasing difference between sensors. The influence of de-synchronization on the attitude estimation should be investigated in further studies. Ths statement was added to the synchronization chapter.
Thank you again for your valuable comments.
Best regards
Author
Reviewer 3 Report (New Reviewer)
The author answered to my inquiries and made the appropriate edits to the paper.
Author Response
Dear Reviewer #3,
thank you for your review.
Best regards
Author
This manuscript is a resubmission of an earlier submission. The following is a list of the peer review reports and author responses from that submission.
Round 1
Reviewer 1 Report
The article presents a low-cost solution in which the drone's flight controller is used as a IMU sensor for human motion measurement.
I have a problem with the review of this work. The author carefully presents the following stages of his reasoning and work, but it is more of a part of report/conference paper/engineering paper than a solid scientific article summarizing a complex data of research work. The author's attention focuses on the construction of the acquisition system itself, while I would expect a demonstration of its utilitarianism in a complex research problem and extensive comparative studies of effectiveness with other currently available solutions.
In recent years, many such non-standard concepts have appeared. I remember when Giuseppe Loianno presented his quadrotor in Ted Talks, in which he used a mobile phone with an IMU to stabilize the quadrotor in his student work at uPenn. Based on this example, we can ask a question, why not use solutions and electronics from mobile phones in this case? Those IMUs are really cheap and precisely nowadays.
In this work, I expected to add more experimental data that would show the advantages of the solution in terms of statistical analyzes of results, as well as comparisons with other devices of this class to show direct pros and cons. Saying that relationship of the cost/effect at the cost of 50 USD per single smart sensor with long-term recorder is excellent, is not enough without direct comparisons in variety of tests and application cases.
Author Response
Dear Reviewer,
the responces are in the attached file.

Reviewer 2 Report
Dear Authors,
The work is interesting mainly because of the economic advantage. But, as part of the reader I have my own reservations as following
1. The time sync that is achieved is not clear. I suggest the authors mention the calculations much more clearly using mathematical tools
2. The results section represents gaits and angles of a few joints and their patterns. This is not expected from the results section. I would rather look at the accuracy with state-of-the-art tech that is available. I encourage the authors show a comparative test with one of the sensors that is mentioned in the related work.
3. Typically the human body movements are captured and represented using quaternions. I think the authors should follow the widely accepted tools to measure the orientation of human body bone-segments.
4. There are tools like Motion-Sphere (https://www.mdpi.com/2076-3417/10/18/6462) which could be used to study the differences and provide the accuracy of a certain human body movement like gait, squat and so on. The authors must use such tools to represent the accuracy of measurement.
5. I would also encourage the authors to use openly available datasets to prove the correctness of the synchronization algorithm that is adopted.
6. There are plethora of multi-sensor time synchronization algorithms that could be adopted. I encourage the authors to explore this area.
7. In my view the results should be presented widely in three perspectives
-> Show Correctness of the synchronization
-> Show Economic advantage
-> Show accuracy with help of either comparative study with state-of-the-art, OR use Goniometer to measure accuracy
NOTE: Its better to use quaternion notations instead of angles.
If the above changes are made, I am sure this work will be a very interesting work.
Author Response
Dear Reviewer
the responces are in the attached file

Round 2
Reviewer 2 Report
Dear Authors,
You have brought in changes in the manuscript and added parts describing the accuracy of the sensor with the state-of-the-art. I still believe that more complex motions could tested for a long duration. Representing range of motions could be interesting.
I could see some outliers in the error in the range of plus or minus 10 degrees. Please discuss why is that happening and are there any solutions.
